# Aetiology and severity of childhood pneumonia in primary care in Malawi: a cohort study

Joe Gallagher ![ORCID],[1,2] Master Chisale ![ORCID],[3] Sudipto Das,[4] Richard J Drew,[5] Nadezhda Glezeva,[1] Dermot Michael Wildes ![ORCID],[1] Cillian De Gascun,[6] Tsung-Shu Joseph Wu ![ORCID],[7,8] Mark T Ledwidge,[1] Chris Watson,[9] On behalf of BIOTOPE team

► Prepublication history and supplemental material for this paper is available online. To view these files, please visit the journal online (http://dx.doi.org/10.1136/bmjopen-2020-046633).

**Correspondence to**
Dr Joe Gallagher;
jgallagher@ucd.ie

## ABSTRACT

**Objective** To determine the aetiology of community acquired pneumonia in children presenting to primary care in Northern Malawi, and to ascertain predictors for identification of children requiring hospitalisation.

**Design** The BIOmarkers TO diagnose PnEumonia study was a prospective cohort study conducted from March to June 2016.

**Setting** Primary care in Northern Malawi.

**Patients** 494 children aged 2–59 months with WHO defined pneumonia.

**Main outcome(s) and measure(s)** Number of children with bacterial infection identified and the sensitivity/specificity of WHO markers of severity for need for hospitalisation.

**Results** 13 (2.6%) children had a bacterium consistent with pneumonia identified. A virus consistent with pneumonia was identified in in 448 (90.7%) of children. 56 children were admitted to hospital and two children died within 30 days. 442 (89.5%) received antibiotic therapy. Eleven children (2.6%) had HIV. WHO severity markers at baseline demonstrated poor sensitivity for the need for hospitalisation with a sensitivity of 0.303 (95% CI 0.188 to 0.441) and a specificity 0.9 (95% CI 0.868 to 0.926). A prediction rule to indicate the need for hospitalisation was developed.

**Conclusions and relevance** The low rate of bacterial infection and high use of antibiotics in the setting of high immunisation rates highlights the changing profile of childhood pneumonia. Similarly, the markers of need for hospitalisation may have changed in the setting of extended immunisation. Further studies are required to examine this.

## Strengths and limitations of this study

⇒ The pneumococcal vaccine was introduced in 2012 in Malawi and this study provides data on the aetiology and severity markers in primary care following the introduction of this vaccine.
⇒ It demonstrates the significant reduction in bacterial infection compared with the previous studies undertaken using similar methods prior to introduction of this vaccine in Africa.
⇒ It also highlights that the severity markers in childhood pneumonia are changing and this is important since these markers are still used in primary care in Africa for the determination of referral to hospital.
⇒ The study was in an urban area with high immunisation rates over one season and the lack of a control group (ie, well children) limited our ability to verify the clinical significance of most viral pathogens and atypical bacteria.

including *Haemophilus influenzae* type B (Hib) and pneumococcal vaccines are likely to have altered the proportion with viral illness. This may also have altered the aetiology, severity and host response in childhood pneumonia. A greater understanding and characterisation of these factors in primary care is required to continue advances in reducing childhood mortality from pneumonia.

Current strategies for the management of childhood pneumonia in low/middle-income countries focus on the identification of the illness and provision of antibiotics, despite the low reported prevalence of bacterial infection.[5] Most studies have focused on the hospitalised population, and yet in developing countries, it is in the community where these patients are most likely to present. Therefore, contemporary estimates of the microbiological causes of pneumonia and predictors of hospitalisation would be of benefit. The use of microbiological techniques is hampered by the high rate of carriage of bacteria and

## INTRODUCTION

Despite recent advances in reducing childhood mortality, pneumonia remains a major cause of morbidity and mortality among children aged under 5 years, with an estimated 935 000 deaths per annum.[1] Allied to this is the growing concern regarding antimicrobial resistance driven by overuse of antibiotics and intercontinental spread of this resistance.[2–4] The increased use of universal immunisation,

BMJ

viruses in children.[6] Finally, WHO markers of severity, which are used to guide referral for hospitalisation in primary care, are mostly based on clinical symptoms alone and advise the use of pulse oximetry to help determine severity if available. However oximeters are rarely available in primary care in countries such as Malawi.[7]

The BIOmarkers TO diagnose PnEumonia (BIOTOPE study) was a prospective cohort study and this paper reports on the aetiology of community acquired pneumonia in children presenting in primary care in Northern Malawi and predictors of children requiring hospitalisation.

## METHODS

From March to June 2016, we recruited children at Mapale Health Centre and the outpatient department of Mzuzu Central Hospital in Mzuzu, Northern Malawi—both of which serve as primary care facilities for this urban area. Shortly after study commencement, the outpatient department of Mzuzu Central Hospital ceased to provide primary care services. As such, 98% of participants were recruited from Mapale Health Centre. Mapale Health Centre currently serves the whole city as a primary health-care facility (population of approximately 220 000) The under-5 population in this setting is characterised by a high burden of infectious disease including acute respiratory infections, diarrhoea and malaria. In 2016, the estimated coverage was 83% for pneumococcal conjugate vaccine and 84% for HiB vaccine in Malawi.[8]

(Further details on site are available in online supplemental file 1).

### Study participants

Children presenting to the primary care facilities were assessed for study enrolment. Children aged 2–59 months were deemed eligible to participate in the study if the main presenting complaint was cough or difficulty breathing associated with tachypnoea (defined as >50 breaths per minute if aged 2–11 months or >40 breaths per minute age if aged 12–59 months) or chest in-drawing. This is the current WHO clinical case definition of pneumonia.[5] Exclusion criteria listed were as follows: those discharged from hospital in the preceding 30 days; those who had completed a course of antibiotics within 14 days of presentation; or those who had received antibiotics prior to clinical assessment for this illness. The study details were discussed with a parent/guardian of any patient who fit our inclusion criteria, who in turn provided informed consent in a written format.

### Study procedures
#### Clinical assessment

In order to record clinical data, a standardised report form was provided to our investigators incorporating demographics, risk factors, and items relevant to a pneumonia presentation. Immunisation records are held by patients in a patient held record known as the 'health passport'. Only those who had a health passport with recorded immunisations were used for purpose of recording immunisation status. Where possible, children were followed up by contacting a caregiver at 7 days and 30 days following initial assessment by telephone. All children had a full blood count and malaria rapid diagnostic test. HIV rapid diagnostic testing was offered to all children aged over 12 months and HIV PCR testing to all children aged 2–11 months.

### Microbiological and other investigations

For all patients, blood samples (8 mL) and pooled naso/oropharyngeal swabs were collected for microbiological investigation on site, and for further serological and molecular analysis in Ireland. A systematic set of investigations were performed for all patients. Blood cultures were processed locally. Blood samples for PCR for *Streptococcus pnuemoniae*, *Staphylococcus aureus* and *H. influenzae* along with naso/oropharyngeal swabs for respiratory viruses and atypical bacteria were processed in Ireland. (Further details in online supplemental file l)

### Definitive diagnosis based on predefined criteria

The final diagnosis (in some cases, diagnoses) was ascertained for each patient using predetermined criteria for both clinical and microbiological assessment. A case was deemed to be 'severe' should the child's condition fulfil WHO criteria for this state. Bacterial pneumonia was diagnosed on a positive blood culture or positive blood PCR for a relevant organism or positive nasopharyngeal swab for *Chlamydia pneumoniae* or *Mycoplasma pneumoniae*. Laboratory data were combined with the predetermined clinical criteria to establish each diagnosis.

### Statistical analyses

The overall aim of this project was to determine blood and urine biomarkers of host response in bacterial pneumonia in a cohort of children age 2–59 months in a single centre in Malawi. Based on the study by D'Acrement *et al*[9] it was estimated that the prevalence of bacterial infection among febrile children with pneumonia would be 20%. The required sample size for our study based on this prevalence was given by the formula $n=4pq/d2$ where p=prevalence q=100 p and d=the precision of the estimate. Assuming a relative precision of 20% and based on previous pilot work on follow-up and sample analysis this resulted in a required sample size of n=490.

Data were entered into Research Electronic Data Capture electronic data capture tools.[10] All statistical analyses were performed using R software (R Foundation for Statistical Computing). Sensitivity and specificity of WHO severity markers for hospitalisation were calculated, with the Clopper-Pearson method used for their confidence intervals. Logistic regression using predefined variables was undertaken to identify a new clinical prediction model and the Area under the curve (AUC) was calculated by area summation, Tjur r-squared estimate and Hanley and McNeil's (1982) SE formula was used for AUC confidence intervals. A simplified clinical prediction rule was

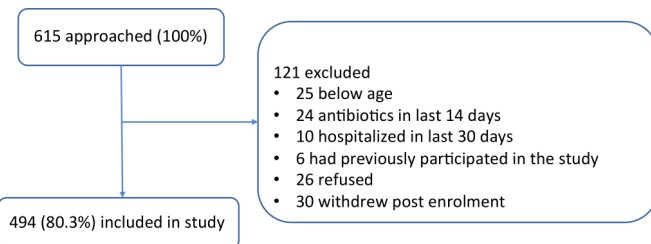

**Figure 1** Participant flow diagram.

developed by selecting independent predictors of hospitalisation measurable from multivariable analyses, allocating a single point for each characteristic and creating a sum score for each patient. We applied receiver operated characteristic analysis at various point thresholds of the score.

## Patient

Patients were not involved in concept design, participant recruitment or study conduct at any stage from inception to completion of this project.

## RESULTS
### Study sample

A total of 615 children were approached for inclusion in this study (figure 1). Demographic and clinical characteristics of the children included in the study are described in table 1. At day 7, 225 caregivers were contactable by phone and 100% of children were alive. At day 30, 195 caregivers were contactable and 2 children had died.

### Immunisations

A total of 317 patients had a patient passport with immunisation record available. A total of 239 of these patients were over the age of 12 months, and 109 were over 24 months of age. Ninety-eight (89.9%) of those aged 24 months or over who had immunisation records available had received full immunisations. Furthermore, of the 56 patients who were hospitalised, 45 had immunisation records available. We age and sex matched this group in a 1:2 ratio with 90 patients who were not-hospitalised and had immunisation records available. The median number of immunisations among the hospitalised cohort was significantly lower than the non-hospitalised cohort (9 (IQR 7–10) vs 10 (IQR 10–10), p=0.003). Children without full complement of vaccinations required by age 1 year were more likely to be hospitalised (RR 5.5, (95% CI 1.88 to 16.1), p=0.008) than those with the correct vaccinations in age-adjusted and sex-adjusted analyses.

### Microbiological analyses

A bacterium consistent with pneumonia was identified in 13 (2.6%) children. In three cases, more than one bacterial cause was identified in the same child. Respiratory viruses were found in 12 of the 13 children considered to have bacterial pneumonia. In 448 (90.7%) of children a respiratory virus alone was identified on testing. Full

details of the microbiological analyses are provided in table 2 and online supplemental file 1. In 33 (6.7%) cases, no organism associated with pneumonia was identified.

### WHO severity criteria and hospitalisation

WHO severity signs and other variables were recorded prospectively. The sensitivity and specificity of these are in table 3.

Using logistic regression, further models were developed with significant improvement in discriminant value. In multivariable modelling, the following seven variables were predictive of hospitalisation: Difficulty breathing, Deep breathing, respiratory rate, age, wheeze, lower chest wall indrawing. The AUC of the fully adjusted model for prediction of hospitalisation was equal to 0.92 (95% CI 0.90 to 0.93, p<0.0001). To create a simplified version of this model, we assigned a score of 1 to each sign or symptom (including categorical treatment of age <2 years and respiratory rate >70 bpm) and generated a simple, 7 point 'DIFFICULTY DRAWING breath' score for each patient, which predicted need for hospitalisation with an AUC of 0.91 (95% CI 0.87 to 0.95, p<0.0001). A score of 0–1 was associated with low need for hospitalisation, 2–3 was intermediate and a score of 4 or greater suggested a high need for hospitalisation. Also, a score of 3 or more had a sensitivity of 87.5% (95% CI 95.9 to 94.8) and a specificity of 84.0% (95% CI 80.2 to 87.4) for hospitalisation (online supplemental file l). This score is predictive of hospitalisation, independent of age, malaria status, HIV status, number of vaccinations, number of people usually sleeping in the same room as the child, presence of chimney for indoor fire, availability of electricity in the home, distance from the nearest health clinic and staff designation of severe pneumonia using WHO Intergrated Management of Childhood Illness (IMCI) criteria. The DIFFICULTY DRAWING breath score was also independently associated with staff designation of severe pneumonia. However, we observed that there was no relationship between staff designation of severe pneumonia using WHO criteria and the decision to admit. This score does require added calculation compared with the WHO severity markers which only require any one of the items and also requires a decision threshold as to the level that requires referral to hospital. However, multicomponent scores are common in clinical practice and we believe that evaluation of such scores and consideration of trials integrated them into electronic decision supports is warranted.[11] The score here also has a high sensitivity compared with the WHO score which has a high specificity. We believe that in primary care a score with a high sensitivity is more appropriate to ensure that false negatives are limited.[12] These aspects will need to be evaluated and validated in future studies.

## DISCUSSION

This study demonstrates that in an area of high immunisation uptake that the prevalence of bacterial pneumonia

**Table 1** Baseline demographic and clinical characteristics

| Demographics | | Total population n=494 | Not hospitalised (n=438) | Hospitalised (n=56) | |
|---|---|---|---|---|---|
| Age (months) | Median (IQR) | 18 (10–30) | 19 (10–32) | 13 (6–21) | <0.001 |
| Age 2–11 months | n (%) | 157/494 (31.8%) | 132/438 (30.1%) | 25/56 (44.6%) | 0.033 |
| Age 12–35 months | n (%) | 252/494 (51.0%) | 224/438 (51.1%) | 28/56 (50.0%) | NS |
| Age 36–60 months | n (%) | 85/494 (17.2%) | 82/438 (18.7%) | 3/56 (5.3%) | 0.013 |
| Male | n (%) | 271/494 (54.8%) | 240/438 (54.8%) | 31/56 (55.4%) | NS |
| Weight for age* > –1 SD | n (%) | 371/481 (77.1%) | 336/438 (76.7%) | 35/56 (62.5%) | 0.032 |
| Weight for age –2 to –1 SD | n (%) | 68/481 (14.1%) | 55/438 (12.6%) | 13/56 (23.2%) | 0.038 |
| Weight for age –3 to –2 SD | n (%) | 34/481 (7.1%) | 27/438 (6.1%) | 7/56 (12.5%) | NS |
| Weight for age < –3 SD | n (%) | 8/481 (1.7%) | 8/438 (1.8%) | 0/56 (0%) | NS |
| Respiratory rate (bpm) | Median (IQR) | 60(54-65) | 59 (52.5–64) | 68(62-72) | <0.0001 |
| Body temperature (°C) | Median (IQR) | 37.4 (36.8–38.2) | 37.4 (36.8–38.1) | 37.8 (37.1–38.6) | 0.015 |
| Antibiotics given | n (%) | 442/466 (94.8%) | 390/411 (94.9%) | 52/55 (94.6%) | NS |
| Antimalarial given | n (%) | 41/441 (9.3%) | 34/392 (8.7%) | 7/49 (14.3%) | NS |
| HIV positive | n (%) | 11/494 (2.2%) | 10/438 (2.2%) | 1/56 (1.8%) | NS |
| Malaria RDT positive | n (%) | 70/485 (14.3%) | 59/438 (13.5%) | 11/56 (19.6%) | NS |
| Clinical Characteristics | | | | | |
| WHO defined severity† | n (%) | 64/494 (12.9%) | 47/438 (10.7%) | 17/56 (30.4%) | <0.001 |
| Fever | n (%) | 470/494 (95.1%) | 417/438 (95.2%) | 53/56 (94.6%) | NS |
| Cough | n (%) | 492/494 (99.6%) | 436/438 (99.5%) | 56/56 (100%) | NS |
| Vomiting | n (%) | 122/494 (24.7%) | 99/438 (22.6%) | 23/56 (41.1%) | 0.005 |
| Diarrhoea | n (%) | 22/494 (4.5%) | 19/438 (4.3%) | 3/56 (5.4%) | NS |
| Abnormal sleepiness | n (%) | 8/494 (1.7%) | 6/438 (1.4%) | 2/56 (3.6%) | NS |
| Coryza | n (%) | 259/494 (52.4%) | 230/438 (52.5%) | 29/56 (51.9%) | NS |
| Sneezing | n (%) | 156/494 (31.6%) | 144/438 (32.9%) | 12/56 (21.4%) | NS |
| Difficulty breathing | n (%) | 184/494 (37.2%) | 138/438 (31.5%) | 46/56 (82.1%) | <0.0001 |
| Deep breathing | n (%) | 53/494 (34.8%) | 26/438 (5.9%) | 27/56 (48.2%) | <0.0001 |
| Wheeze | n (%) | 36/494 (7.3%) | 25/438 (5.7%) | 11/56 (19.6%) | 0.001 |
| In-drawing lower chest wall | n (%) | 172/494 (34.8%) | 128/438 (29.2%) | 44/56 (78.6%) | <0.0001 |
| Nasal flaring | n (%) | 216/494 (43.7%) | 175/438 (40.0%) | 41/56 (73.2%) | <0.0001 |
| Grunting | n (%) | 28/494 (5.6%) | 14/438 (3.2%) | 14/56 (25.0%) | <0.0001 |
| Head nodding | n (%) | 17/494 (3.4%) | 10/438 (2.2%) | 7/56 (12.5%) | 0.001 |

*Weight was not recorded in 11 children and 3 children were oedematous.
†Describes 64 children with any one of severe malnutrition (weight for age <3 SD, n=8), unable to feed/drink (n=50), stridor (n=9), lethargy (n=0), unconsciousness (n=0), vomiting everything (n=0), convulsions (n=0).
NS, not significant; RDT, rapid diagnostic test.

in children presenting in primary care with clinically defined pneumonia is low. The introduction of both HiB and pneumococcal conjugate vaccines has been shown to reduce the incidence of severe bacterial infections and hospitalisation among children.[13 14] This data are consistent with the high uptake of immunisations in our cohort. A recent study has also shown that, in Malawi, the number of children with non-severe, fast-breathing pneumonia that needed amoxicillin treatment, for one child to benefit was 33 which is supportive of our findings also.[15] The number of children with bacterial pneumonia has declined significantly since a previous study in the community in Eastern Africa performed prior to the introduction of pneumococcal vaccine although, this used different microbiological techniques to assign diagnoses.[9] The PERCH study which examined the causes of severe childhood pneumonia requiring hospital admission has also shown a predominantly viral aetiology even in this severe subset.[16] Overall, very few studies have been conducted in primary care in the era prior to the introduction of pneumococcal vaccine.[15–19] Given that most children with pneumonia will present initially to primary

**Table 2** Microbiological analyses

| Bacterial* | | Total population n=488 | Not hospitalised (n=432) | Hospitalised (n=56) | |
|---|---|---|---|---|---|
| *Streptococcus pneumoniae* | n (%) | 4/494 (0.8%) | 4/438 (0.9%) | 0/56 (0 %) | NS |
| *Staphylococcus aureus* | n (%) | 2/494 (0.4%) | 2/438 (0.5%) | 0/56 (0%) | NS |
| *Haemophilus* spp. | n (%) | 2/494 (0.4%) | 2/438 (0.5%) | 0/56 (0%) | NS |
| *Mycoplasma pneumoniae* | n (%) | 3/494 (0.6%) | 3/438 (0.7%) | 0/56 (0%) | NS |
| *Chlamydia pneumoniae* | (%) | 2/494 (0.4%) | 2/438 (0.5%) | 0/56 (0%) | NS |
| Any bacterial infection | n (%) | 13/494 (2.6%) | 13/438 (3.0%) | 0/56 (0%) | NS |
| Viral† | | | | | |
| Adenovirus | n (%) | 82/494 (16.6%) | 71/438 (16.2%) | 11/56 (19.6%) | NS |
| Bocavirus | n (%) | 100/494 (20.2%) | 92/438 (21%) | 8/56 (20.5%) | NS |
| CoV HKU | n (%) | 0/494 (0%) | 0/438 (0%) | 0/56 (0%) | NS |
| CoV NL63 | n (%) | 1/494 (0.2%) | 1/438 (0.2%) | 0/56 (0%) | NS |
| CoV 229E | n (%) | 1/494 (0.2%) | 1/438 (0.2%) | 0/56 (0%) | NS |
| CoV OC43 | n (%) | 1/494 (0.2%) | 1/438 (0.2%) | 0/56 (0%) | NS |
| Influenza A | n (%) | 76/494 (15.4%) | 74/438 (16.9%) | 2/56 (15.6%) | NS |
| Influenza AH1 | n (%) | 64/494 (13%) | 63/438 (14.4%) | 1/56 (1.8%) | 0.001 |
| Influenza AH3 | n (%) | 6/494 (1.2%) | 6/438 (1.4%) | 0/56 (0%) | NS |
| Influenza B | n (%) | 9/494 (1.8%) | 8/438 (1.8%) | 1/56 (1.8%) | NS |
| HMPV | n (%) | 73/494 (14.7%) | 63/438 (14.4%) | 10/56 (15.0%) | NS |
| Parainfluenza 1 | n (%) | 11/494 (2.2%) | 10/438 (2.3%) | 1/56 (1.8%) | NS |
| Parainfluenza 2 | n (%) | 10/494 (2.0%) | 10/438 (2.3%) | 0/56 (0%) | NS |
| Parainfluenza 3 | n (%) | 4/494 (0.8%) | 3/438 (0.7%) | 1/56 (1.8%) | NS |
| Parainfluenza 4 | n (%) | 17/494 (3.4%) | 15/438 (3.4%) | 2/56 (3.6%) | NS |
| RSV A | n (%) | 164/494 (33.2%) | 144/438 (32.9%) | 20/56 (35.7%) | NS |
| RSV B | n (%) | 50/488 (10.1%) | 44/438 (10%) | 6/56 (10.7%) | NS |
| Rhino/enterovirus | n (%) | 295/494 (59.7%) | 263/438 (60%) | 32/56 (57.1%) | NS |
| Any virus | n (%) | 940/488448/494 (93.1%) | 409/432 (94.7%) | 51/56 (91.1%) | NS |

*Two other bacterial species were identified in the patients who were not hospitalised, but were not clinically associated with pneumonia (*Escherichia coli* and *Shigella*).
†In six cases, it was not possible to obtain a valid PCR run.
CoV, coronavirus; HMPV, Human Metapneumovirus; NS, not significant; RSV, Respiratory Synctial Virus.

**Table 3** Sensitivity and specificity of WHO markers of severe pneumonia for hospitalisation

| Variable | Sensitivity (95% CI) | Specificity (95% CI) |
|---|---|---|
| Severe pneumonia | 0.304 (0.188 to 0.441) | 0.9 (0.868 to 0.926) |
| Vomiting everything | – | 0.996 (0.984 to 0.999) |
| Convulsions | – | 1 (0.992 to 1.000) |
| Lethargy | – | 1 (0.992 to 1.000) |
| Unconsciousness | – | 1 (0.992 to 1.000) |
| Malnourished | – | 0.987 (0.971 to 0.995) |
| Unable to feed | 0.214 (0.116 to 0.344) | 0.916 (0.886 to 0.940) |
| Stridor | 0.089 (0.030 to 0.196) | 0.991 (0.977 to 0.998) |
| Any sign | 0.179 (0.089 to 0.304) | 0.938 (0.911 to 0.958) |

care, there is a clear need for further evaluation of both the aetiology and severity profile of childhood pneumonia in this setting.

The markers of severity (defined as hospitalisation) in the BIOTOPE study are different from that outlined in current WHO guidelines even those these are the guidelines used to guide who should be referred to hospital. Our study shows that WHO severity criteria are present in a minority (30.4%) of children hospitalised, consistent with recent work showing that 39% of fatal cases of pneumonia were defined as having non-severe pneumonia requiring only home treatment by the 2013 revision.[18] The fact that the WHO criteria would have discharged these children with oral antibiotics highlights the need for new markers of severity, as these guidelines are used to guide primary care workers in their decision making regarding escalation of care. Our study highlights the changing pattern of markers of severity for hospitalisation and generates new markers of severity that could be used in the community. We present a new 7-point clinical prediction rule (DIFFICULTY DRAWING Breath score) based on independent predictors of severity with a high predictive power for severity, defined as need for hospitalisation. This will need to be evaluated in future studies.

This study involves an urban centre in an area of high immunisation and results may differ in different settings, particularly in locations with low immunisation uptake rates or where healthcare may not be easily accessed, for example, remote areas. Infection rates for both respiratory viruses and malaria are subject to significant seasonal variation, as such, had the study been conducted during an alternative time period or in an alternative geographical location, the results may have differed significantly. However, given the study site was representative of Malawi and the Mzuzu region as a whole in terms of immunisation rates, we believe our findings may be generalised and applied to Malawi as a nation and (to a certain extent) other nations of sub-Saharan Africa. More studies are required in different epidemiological settings to generate further data. The lack of a control group (ie, well children) limited our ability to verify the clinical significance of most viral pathogens and atypical bacteria. However, all children were included based on a clinical definition of pneumonia and therefore had pre-existing illness signs. Our rate of identification of a possible pathogen was high and this likely reflects the use of molecular techniques. The lack of chest radiography may be seen as a limitation but previous studies have shown it is not predictive of aetiology and does not improve outcomes.[19] Also, plain film radiographs would not be routinely available in primary care. The use of hospitalisation as a marker of severity may be seen as a limitation as hospitalisation may vary based on hospital practice and health seeking behaviour. However, this study was based in primary care and hospitalisation is a useful standard to guide those who should be referred from primary care. Larger studies to evaluate and validate scores for markers such as mortality are required

Our findings indicate that among preschool-aged children in a region with high rates of immunisation, a high percentage of WHO clinically defined pneumonia are likely to be of viral origin. It is known that respiratory viruses, including rhinoviruses, can cause severe lower respiratory tract disease, particularly in infants and young children.[20 21] There is also a change in the clinical markers of severe illness, as defined by hospitalisation. The current WHO markers of severity have a low sensitivity for hospitalisation. This has important implications for the management of childhood pneumonia in the community and may reflect the change in clinical signs in the era of high immunisation uptake.

## CONCLUSION

In Mzuzu, Malawi, the vast majority of children with a WHO clinical definition of pneumonia had a viral illness and thus did not require an antibiotic. Current guidelines suggest all of these children should receive antibiotics. This subjects the community to accelerated development of antimicrobial resistance. The current definition of severe clinical signs used by the WHO has a poor sensitivity and this is potentially attributable to the high uptake of immunisations against respiratory bacteria in this cohort, with a corresponding evolution of disease presentation. Further study is warranted to determine if a revision of current WHO guidelines is required, thereby ensuring children are appropriately are referred to hospital. The score presented here provides a practical alternative, should it be more widely validated.

**Author affiliations**
[1]gHealth Research Group, University College Dublin College of Health Sciences, Dublin, Ireland
[2]School of Medicine, University of Limerick, Limerick, Ireland
[3]Biological Science Department, Faculty of Science Technology and Innovations, Mzuzu University, Mzuzu, Malawi
[4]School of Pharmacy and Biomolecular Sciences, Royal College of Surgeons in Ireland, Dublin, Ireland
[5]Irish Meningitis and Sepsis Reference Laboratory, Dublin, Ireland
[6]National Virus Reference Laboratory, Dublin, Ireland
[7]Luke International Norway, Mzuzu, Malawi
[8]Overseas Mission Department, Pingtung Christian Hospital, Pingtung, Taiwan
[9]Queen's University Belfast Wellcome-Wolfson Institute for Experimental Medicine, Belfast, UK

**Correction notice** This article has been corrected since it was published. Surname of author 'Nadezhda Glezeva' has been corrected.

**Contributors** Contributors JG and CW conceived the paper, assisted with study design and secured funding. MTL performed data analysis, interpretation and drafted the manuscript. SD, RJD, NG, CDG, MC, DMW and T-SJW assisted with study design, analysis and interpretation of data. All authors contributed to manuscript development, edited for critical content, and have approved the final version.

**Funding** This work was conducted with the generous support of the Bill & Melinda Gates Foundation – Investment ID: OPP1139557.

**Competing interests** None declared.

**Patient consent for publication** Not required.

**Ethics approval** The protocol and related documents were approved by the National Health Science Research Committee of Malawi and the Mzuzu Central

Hospital Research Committee. Patients received timely treatment and supportive care as required. The study was conducted in compliance with the ethical standards as outlined by two ethics committees and was aligned with the values outlined by the Declaration of Helsinki. (National Health Research Committee of Malawi Ethics Approval #15/11/1532).

**Provenance and peer review** Not commissioned; externally peer reviewed.

**Data availability statement** Data are available on reasonable request. Study data pertaining to the BIOTOPE Cohort, may be made available on reasonable request from the corresponding author.

**ORCID iDs**
Joe Gallagher http://orcid.org/0000-0002-5564-2890
Master Chisale http://orcid.org/0000-0002-8301-6184
Dermot Michael Wildes http://orcid.org/0000-0001-6281-5713
Tsung-Shu Joseph Wu http://orcid.org/0000-0001-6155-9340

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
