## [Reviewer comments · BMJ Open]

ARTICLE DETAILS

TITLE (PROVISIONAL)	Aetiology and Severity of Childhood Pneumonia in Primary Care in Malawi: A Cohort Study
AUTHORS	Gallagher, Joe; Chisale, Master; Das, Sudipto; Drew, Richard; Gleseva, Nadezhda; Wildes, Dermot; De Gascun, Cillian; Wu, Tsung-Shu; Ledwidge, Mark; Watson, Chris

VERSION 1 – REVIEW

REVIEWER	Onwuchekwa, Chukwuemeka Institute of Tropical Medicine
REVIEW RETURNED	28-Dec-2020

GENERAL COMMENTS	Thank you for the opportunity to review this manuscript on "Aetiology and Severity of Childhood Pneumonia in Primary Care in Malawi". I have read your manuscript and find it to be well written and addresses an important topic. Below I make a few general and more specific comments on the manuscript. General comments: Introduction section: The background was fairly well written, however, there is a slight incoherence in this section. This is detailed further below. Methodology: The method section is lacking in depth, I recommend more detailed description of the study design, population and study setting. This study reads like a case series, but I wonder why a follow-up period was required? Also, the laboratory methods need to be more detailed. Result: The results are overall well presented. Some of the results presented were not coherent with the study aims and methodology. Discussion: No major changes recommended. Specific comments: 1. Page 5 line 43 - 48: Is this statement related to febrile illnesses in general or pneumonia? the first paragraph of the introduction talk about pneumonia while the subsequently you refer to febrile illness2. Page 5, line 53: This statement will require a citation3. Page 5, line 55 - 57: Does this relate to use of antibiotics in pneumonia treatment? because I disagree that the rise in malaria RDT is likely to have increased use of antibiotics in treatment of children presenting with symptoms consistent with WHO pneumonia case definition. The paper cited here looked at persons who had an RDT test, not necessarily those with pneumonia symptoms.4. page 6, line 18-26:
--

	the current WHO markers of severity combined clinical symptoms, signs and oxygen saturation measured by pulse oxymeter 5. Page 6, line 50 - 57: I would recommend more information on the study site and population. For example, more of the health facilities and the services available. Also, a little more on the population these facilities serves (hence the population from which the study group is drawn) 6. page 8 line 42 - 50: I recommend more information on how samples were processed. For instance for blood cultures what procedure was used? for the PCR, which PCR method was used? 7. Page 10, line 21 - 23: I need to understand why there was a follow-up period in this study? 8. Page 10, sub-heading (immunisations): I do not see the importance of this result presented. I recommend comparing immunization rates between hospitalized and non-hospitalized cases (as bacterial pneumonia are more likely to be severe and hence patients hospitalized). 9. Page 10, line 47 - 59: it is not clear the source (i.e sample source) from which these bacteria/viruses were isolated? (Blood?NPS or both?) also the method of identification should be clarified. (PCR, culture, both) 10. Page 10, line 54: it will be difficult to talk of causality based on NP swabs alone as this cannot discriminate carriage (therefore is not specific for disease). I see this is touched on in the discussion section. Consider changing the wording of this sentence. 11. Page 12, line 18: This statement is likely linked to reference No. 10 (Hammit et al) and not No. 6. 12. Page 12, line 45 - 49: The markers of severity you proposed include symptoms/features that do not discriminate severe pneumonia from other illnesses (eg stridor, head nodding, convulsion etc).
--	--

REVIEWER	Edem, Bassey MRC Unit The Gambia at LSHTM
REVIEW RETURNED	10-Jan-2021

GENERAL COMMENTS	The Authors have demonstrated evidence of the changing aetiology of childhood pneumonia in Malawi post introduction of conjugate vaccination and also developed a tool for predicting hospitalization that can be validated in larger studies. However, it is not mentioned at any point how the sample size for this study was estimated. This limits the external validity of the study findings as a whole and speaks to the generalizability of the study findings (lines 47-50). As a background to the study, it would be useful to provide coverage estimates for Malawi for PCV and HiB routine immunization. Line 24: AUC in full
---

VERSION 1 – AUTHOR RESPONSE

Reviewer: 1

Dr. Chukwuemeka Onwuchekwa, Institute of Tropical Medicine

We thank Dr Onwechekwa for his general comments which were very helpful and have taken these into account in the response to the specific areas below also

Specific comments:

1. Page 5 line 43 - 48:

Is this statement related to febrile illnesses in general or pneumonia? the first paragraph of the introduction talk about pneumonia while the subsequently you refer to febrile illness

We agree that this is confusing and have now rationalised the paragraph to focus on pneumonia only Current strategies for the management of childhood pneumonia in developing countries focus on the identification of the illness and provision of antibiotics, despite the low reported prevalence of bacterial infection⁶⁻⁹. Most studies have focused on the hospitalised population, and yet in developing countries, it is in the community where these patients are most likely to present. Therefore, contemporary estimates of the microbiological causes of pneumonia and predictors of hospitalization would be of benefit. The use of microbiological techniques is hampered by the high rate of carriage of bacteria and viruses in children¹⁰. Finally, WHO markers of severity, which are used to guide referral for hospitalization in primary care, are mostly based on clinical symptoms alone and advise the use of pulse oximetry to help determine severity if available. However oximeters are rarely available in primary care in countries such as Malawi¹¹.

2. Page 5, line 53:

This statement will require a citation

We have added a reference to the WHO guidelines on management of childhood pneumonia to highlight the provision of antibiotics is recommended for all patients with WHO defined pneumonia.

3. Page 5, line 55 - 57:

Does this relate to use of antibiotics in pneumonia treatment? because I disagree that the rise in malaria RDT is likely to have increased use of antibiotics in treatment of children presenting with symptoms consistent with WHO pneumonia case definition. The paper cited here looked at persons who had an RDT test, not necessarily those with pneumonia symptoms.

We agree that this is not clear and agree with Prof Onwuchekwa's point that this relates to febrile illness in general and not to pneumonia in particular. We have removed this line from the text

4. page 6, line 18-26:

The current WHO markers of severity combined clinical symptoms, signs and oxygen saturation measured by pulse oximeter

We have clarified this statement below

Finally, WHO markers of severity, which are used to guide referral for hospitalization in primary care, are mostly based on clinical symptoms alone and advise the use of pulse oximetry to help determine severity if available. However oximeters are rarely available in primary care in countries such as Malawi¹¹

5. Page 6, line 50 - 57:

I would recommend more information on the study site and population. For example, more of the

health facilities and the services available. Also, a little more on the population these facilities serves (hence the population from which the study group is drawn)

We have added further details to the main manuscript and a supplemental file due to space considerations. The detail in the main manuscript is

“Mapale Health Centre currently serves the whole city as a primary health care facility (population of approximately 220,000) The under five population in this setting is characterised by a high burden of infectious disease including acute respiratory infections (ARI), diarrhoea, and malaria.

In 2016 the estimated coverage was 83% for pneumococcal conjugate vaccine and 84% for Haemophilus influenza type B vaccine in Malawi,¹² (Further details on site are available in the supplemental file)”

The supplemental file contains

Mzuzu is one of the four cities in Malawi, with a population of 221,272 (National Statistics Office, 2018). Mzuzu health centre also known as Mapale or Mzimba North Health centre is situated at the City centre of Mzuzu city while Mzuzu Central Hospital is situated at around 4 km Northwards of the city. The health centre currently serves almost the whole city as a primary health care point while the central hospital is regional referral tertiary hospital serving six districts in the northern region, over hundred health centres and serving a population of over 1.9 million people. It also serves as a secondary healthcare facility for the local (Mzuzu city) population. The majority (98.0%) of our participants were from Mapale health centre. The Central hospital initially served as a primary healthcare facility also but prior to commencement of the study the local health authority decided to maintain it as a secondary care facility only. Therefore it introduced a “bypass fee” where patients who attended the hospital clinics without being referred from a health centre/district hospital were charged MK 1,500 (approximately 2 Euros). This led to Mapale becoming the main site of recruitment due to the focus on recruiting patients from primary care and before they had received any treatment.

UNICEF data from 2013 reports an under five mortality rate of 68 per 1,000 live births and under one mortality rate of 44 per 1,000 live births in Malawi. In 2016 the estimated coverage was 83% for pneumococcal conjugate vaccine and 84% for Haemophilus influenza type B vaccine in Malawi,¹¹

HIV prevalence in antenatal clinics in the Mzimba district (where Mzuzu is located) among those who accepted testing was 9.1%¹. A study in 2011 in Malawi showed that of 5,068 samples from infants <3 months of age , 764 were ELISA positive indicating 15.1% (14.1–16.1%) of mothers were HIV-infected and passed antibodies to their infant. Sixty-five of the ELISA-positive samples tested positive by DNA PCR, indicating a vertical transmission rate of 8.5% (6.6–10.7%). Survey data indicates 64.8% of HIV-infected mothers and 46.9% of HIV-exposed infant received some form of antiretroviral prophylaxis².

6. page 8 line 42 - 50:

I recommend more information on how samples were processed. For instance for blood cultures what procedure was used? for the PCR, which PCR method was used?

We have added further details to the supplemental file and outline them below also Blood cultures
The blood culture samples were inoculated into the Bactec Peds Plus/F Culture vial (Becton Dickinson Diagnostic Instrument Systems). These bottles were incubated in an automated blood culture Bactec™ 9050 which was set at protocol limits of maximum of 7 days to declare the sample negative if no growth has been detected. All positive blood cultures were further subcultured on to agar plates and identified using traditional phenotypic methods (Blood agar, MacConkey and Chocolate agar plate) and Vitek 2 microbial detection system (Biomerieux)

Bacterial PCR

The Masterpure™ complete DNA and RNA purification kit as well as QIA Symphony Automated DNA extraction system [manufacturer. Country] which is an automated method was used to extract DNA and RNA. The extracted DNA was run on AB7500 instruments as per previously detailed methods^[1] (see references below)

Viral RT-PCR

Viral RT-PCR was performed on nasopharyngeal and oropharyngeal swabs using a Luminex® MAGPIX® instrument [manufacturer, country] with xPONENT® and SYNCT™ software, in line with manufacturer's instructions. The panel used was NxTAG respiratory pathogen. The pathogens targeted in this assay were; Influenza A, Influenza A H1, Influenza B, Influenza A H3, Respiratory Syncytial Virus A, Respiratory Syncytial Virus B, Coronavirus 229E, Coronavirus OC43, Coronavirus NL63, Coronavirus HKU1, Human Metapneumovirus, Rhinovirus/Enterovirus Adenovirus, Parainfluenza 1, Parainfluenza 2, Parainfluenza 3, Parainfluenza 4, Human Bocavirus, Chlamydia pneumoniae, Mycoplasma pneumoniae, Legionella pneumophila

7. Page 10, line 21 - 23:

I need to understand why there was a follow-up period in this study?

This study was conducted in primary care clinics. Patients who required admission were referred to hospital in a different location and were not admitted to the clinic where the study was undertaken. In order to ensure that any patients who may have went home and were subsequently presented to another facility or went directly to hospital were identified we subsequently tried to contact caregivers to identify those patients who were admitted to hospital following their first presentation by mobile phone follow up.

8. Page 10, sub-heading (immunisations):

I do not see the importance of this result presented. I recommend comparing immunization rates between hospitalized and non-hospitalized cases (as bacterial pneumonia are more likely to be severe and hence patients hospitalized).

We agree that this is a useful analysis and we have included this now in the revised draft. In order to take account of age differences we also undertook further analysis also (text in manuscript below) Furthermore, of the 56 patients who were hospitalised, 45 had immunization records available. We age and sex matched this group in a 1:2 ratio with 90 patients who were not-hospitalised and had immunization records available (Supplemental file). The median number of immunisations amongst the hospitalized cohort was significantly lower than the non-hospitalised cohort (9 [IQR 7,10] vs. 10 [IQR10,10], $p=0.003$). Children without full complement of vaccinations required by age one year were more likely to be hospitalised (RR 5.5, [95% CI: 1.88-16.1], $p=0.008$) than those with the correct vaccinations in age and sex adjusted analyses.

9. Page 10, line 47 - 59:

it is not clear the source (i.e sample source) from which these bacteria/viruses were isolated? (Blood?NPS or both?)

also the method of identification should be clarified. (PCR, culture, both)

We have now included a supplemental file which lists the source of these and the key characteristics for each child (Table S1). We have also included details on the microbiology results by month when samples obtained and by age of child

10. Page 10, line 54:

it will be difficult to talk of causality based on NP swabs alone as this cannot discriminate carriage

(therefore is not specific for disease). I see this is touched on in the discussion section. Consider changing the wording of this sentence.

We have changed this wording to remove an implication of causation

“In 448 (90.7%) of children a respiratory virus alone was identified on testing.”

11. Page 12, line 18:

This statement is likely linked to reference No. 10 (Hammit et al) and not No. 6.

We agree that reference 10 (Hammit et al) is a relevant analysis. However D’Acremont et al also undertook an analysis of bacterial infection in pneumonia prior to introduction of pneumococcal vaccine and this reference refers to this.

12. Page 12, line 45 - 49:

The markers of severity you proposed include symptoms/features that do not discriminate severe pneumonia from other illnesses (eg stridor, head nodding, convulsion etc).

We agree that this is not clear. These markers of stridor, head nodding, convulsion etc are based on current WHO clinical symptoms and signs severity markers. The features in the model proposed in this paper included Difficulty breathing, Deep breathing, Respiratory rate, Age, Wheeze, Lower chest wall indrawing. We have included further details on this in the supplemental file and also highlighted this in the main manuscript

Reviewer: 2

Dr. Basse Edem, MRC Unit The Gambia at LSHTM

Comments to the Author:

It is not mentioned at any point how the sample size for this study was estimated. This limits the external validity of the study findings as a whole and speaks to the generalizability of the study findings (lines 47-50).

We have now included the details below in the statistical analysis section:

The overall aim of this project was to determine blood and urine biomarkers of host response in bacterial pneumonia in a cohort of children age 2- 59 months in a single centre in Malawi. Based on the study by D’Acremont⁶ et al it was estimated that the prevalence of bacterial infection among febrile children with pneumonia would be 20%. The required sample size for our study based on this prevalence was given by the formula $n = 4pq/d^2$ where p = prevalence $q = 100-p$ and d = the precision of the estimate. Assuming a relative precision of 20% and based on previous pilot work on follow up and sample analysis this resulted in a required sample size of $n = 490$.

As a background to the study, it would be useful to provide coverage estimates for Malawi for PCV and HiB routine immunization.

We have added the following text:

In 2016 the estimated coverage was 83% for pneumococcal conjugate vaccine and 84% for Haemophilus influenza type B vaccine in Malawi,¹¹

Line 24: AUC in full

We have added the full text Area under the Curve (AUC)

[1] Meyler KL, Meehan M, Bennett D, Cunney R, Cafferkey M. Development of a diagnostic real-time polymerase chain reaction assay for the detection of invasive Haemophilus influenzae in clinical samples. Diagn Microbiol Infect Dis. 2012 Dec;74(4):356-62. doi: 10.1016/j.diagmicrobio.2012.08.018. Epub 2012 Sep 25. PMID: 23017260.

3: Murphy J, O' Rourke S, Corcoran M, O' Sullivan N, Cunney R, Drew R. Evaluation of the Clinical Utility of a Real-time PCR Assay for the Diagnosis of Streptococcus pneumoniae Bacteremia in Children: A Retrospective Diagnostic Accuracy Study. Pediatr Infect Dis J. 2018 Feb;37(2):153-156. doi: 10.1097/INF.0000000000001772. PMID: 29076932.

VERSION 2 – REVIEW

REVIEWER	Onwuchekwa, Chukwuemeka Institute of Tropical Medicine
REVIEW RETURNED	14-Apr-2021

GENERAL COMMENTS	Thank you for the revised manuscript, and for addressing all my comments. I do not have any specific comment, rather I have a more general comment regarding your proposed severity score. I think your severity score in itself is interesting, though it does not appear to add much to the current WHO integrated management of childhood illness (IMCI) criteria for pneumonia severity. Importantly, by including a severity grade, your proposed severity score now sets a higher threshold for definition of severity than the IMCI guide (which defines a child as having severe pneumonia if they have at least one sign or symptom of severity). I am also concerned about how you attempted to validate your scoring system, using "hospitalisation" as a sort of reference for severity. This makes it a bit problematic because in practice, the decision to admit a child with pneumonia is not based entirely on the severity of their symptoms. Specifically, in line 15 - 21 of page 13, you report the sensitivity and specificity of the scoring system against hospitalizations. I would suggest reporting on associations between scores and hospitalization probability, and report sensitivity and specificity against a more widely accepted measure of severity like the IMCI criteria.
--

VERSION 2 – AUTHOR RESPONSE

I do not have any specific comment, rather I have a more general comment regarding your proposed severity score. I think your severity score in itself is interesting, though it does not appear to add much to the current WHO integrated management of childhood illness (IMCI) criteria for pneumonia severity. Importantly, by including a severity grade, your proposed severity score now sets a higher threshold for definition of severity than the IMCI guide (which defines a child as having severe pneumonia if they have at least one sign or symptom of severity).

These are useful comments to refine the discussion. We accept that the use of a severity score rather than a single item score such as WHO may make calculation more difficult and requires development

of a threshold at which referral is indicated. However there are a limited number of items and such scores are widely used in clinical practice already^[1]. We also believe that the score does add value compared to the WHO score. In our study the WHO signs had high specificity but poor sensitivity. In primary care we believe that a high sensitivity score is required to ensure that false negatives are limited and that children are referred to hospital and that a high sensitivity score is more appropriate in this setting^[2] It also concurs with the concerns raised in the study by Agweyu et al.^[3] showing that 39% of fatal cases of pneumonia were defined as having non-severe pneumonia requiring only home treatment by the 2013 WHO revision

We have added the following to the text to

This score is predictive of hospitalisation, independent of age, malaria status, HIV status, number of vaccinations, number of people usually sleeping in the same room as the child, presence of chimney for indoor fire, availability of electricity in the home, distance from the nearest health clinic and staff designation of severe pneumonia using WHO IMCI criteria. The DIFFICULTY DRAWING breath score was also independently associated with staff designation of severe pneumonia. However, we observed that there was no relationship between staff designation of severe pneumonia using WHO criteria and the decision to admit. This score does require added calculation compared to the WHO severity markers which only require any one of the items and also requires a decision threshold as to the level that requires referral to hospital. However multicomponent scores are common in clinical practice and we believe that evaluation of such scores and consideration of trials integrated them into electronic decision supports is warranted. The score here also has a high sensitivity compared to the WHO score which has a high specificity. We believe that in primary care a score with a high sensitivity is more appropriate to ensure that false negatives are limited. These aspects will need to be evaluated and validated in future studies.

I am also concerned about how you attempted to validate your scoring system, using "hospitalisation" as a sort of reference for severity. This makes it a bit problematic because in practice, the decision to admit a child with pneumonia is not based entirely on the severity of their symptoms. Specifically, in line 15 - 21 of page 13, you report the sensitivity and specificity of the scoring system against hospitalizations. I would suggest reporting on associations between scores and hospitalization probability, and report sensitivity and specificity against a more widely accepted measure of severity like the IMCI criteria.

We accept that hospitalisation is a subjective standard and may vary based on hospital practice and health seeking behaviour. However since this study is based in primary care we believe that hospitalisation is a useful measure of severity. WHO markers of severity are used to guide who should be referred to hospital from primary care also

We have added the following piece to our limitations section

"The use of hospitalisation as a marker of severity may be seen as a limitation as hospitalisation may vary based on hospital practice and health seeking behaviour. However this study was based in primary care and hospitalisation is a useful standard to guide those who should be referred from

primary care. Larger studies to evaluate and validate scores for markers such as mortality are required”

We have also undertaken further analysis to control for co-variates which is presented in the supplemental file for both hospitalisation and WHO defined severe pneumonia

S3: Markers of severity

Figure S1 Difficulty DRAWING Breath score and risk of hospitalisation

DIFFICULTY DRAWING breath score = 1 for each of:

DIFFICULTY breathing

Deep breathing

Respiratory Rate > 70 bpm

Age <2 years

Wheeze

INDrawing lower chest wall

Grunting

Area under the ROC curve

Area 0.8918

Std. Error 0.02002

95% confidence interval 0.8525 to 0.9310

P value < 0.0001

Data

Controls (Not Hospitalised) 438

Patients (Hospitalised) 56

Missing Controls 0

Missing Patients 0

[1] The DD Score (strongly, positively associated) and previous number of vaccines (modestly, negatively associated) are the only independent predictors of hospitalisation in the model with the following covariates included:

- Staff designation of severe pneumonia (IMCI criteria)
- Age
- HIV status (immunosuppression)
- Completed WHO vaccination schedule
- Number sleeping in same room as child (overcrowding)
- Chimney present for indoor fire (indoor smoke)
- Electricity present (marker of socioeconomic status)
- Malaria co-infection (disease severity)
- Distance to nearest clinic (marker of healthcare access)

For each unit increase in the DD Score there is a 10.6% (95% CI 7.9-13.4) increase in the likelihood of severe disease as defined by hospitalisation ($p < 0.0001$)

[2] The DD Score is the only strong, independent predictor of severe pneumonia as designated by the staff in a similar model with the following covariates included:

- Hospitalisation
- Age
- HIV status (immunosuppression)
- Completed WHO vaccination schedule
- Number sleeping in same room as child (overcrowding)
- Chimney present for indoor fire (indoor smoke)
- Electricity present (marker of socioeconomic status)
- Malaria co –infection (disease severity)
- Distance to nearest clinic (marker of healthcare access)

For each unit increase in the DD Score there is a 6.0% (95% CI 2.5-9.6) increase in the likelihood of severe disease as defined by WHO criteria ($p < 0.001$)

Area under the ROC curve

Area 0.69

95% confidence interval 0.62 to 0.76

P value < 0.001

[1] Keogh C, Wallace E, O'Brien KK, Galvin R, Smith SM, Lewis C, Cummins A, Cousins G, Dimitrov BD, Fahey T. Developing an international register of clinical prediction rules for use in primary care: a descriptive analysis. *Ann Fam Med*. 2014 Jul;12(4):359-66. doi: 10.1370/afm.1640. PMID: 25024245; PMCID: PMC4096474.

[2] M.S. Abers, D.M. Musher, Clinical prediction rules in community-acquired pneumonia: lies, damn lies and statistics, *QJM: An International Journal of Medicine*, Volume 107, Issue 7, July 2014, Pages 595–596, <https://doi.org/10.1093/qjmed/hcu096>

[3] Agweyu A, Lilford RJ, English M, Clinical Information Network Author G. Appropriateness of clinical severity classification of new WHO childhood pneumonia guidance: a multi-hospital, retrospective, cohort study. *The Lancet Global health*. Jan 2018;6(1):e74-e83. doi:10.1016/S2214-109X(17)30448-5